# Cutting Force When Machining Hardened Steel and the Surface Roughness Achieved

**Karel Osička *** , **Jan Zouhar** , **Petra Sliwková** and **Josef Chladil**

Faculty of Mechanical Engineering, Brno University of Technology, Technická 2996/2, 61602 Brno, Czech Republic
* Correspondence: osicka@fme.vutbr.cz

**Abstract:** This article deals primarily with the problem of determining the cutting force when machining hardened steels. For this study, the steel used was 100 Cr6, number 1.3505. The secondary aspects of the study focused on the evaluation of the surface quality of machined samples and the recommendation of cutting conditions. A wide variety of components are used in engineering, the final heat treatment of which is hardening. These components are usually critical in a particular product. The quality of these components determines the correct functioning of the entire body of technical equipment, and ultimately, its service life. In our study, these are the core parts of thrust bearings, specifically the rolling elements. The subject of this experiment involves machining these components in the hardened state with cubic boron nitride tools and the continuous measurement of the cutting force using a dynamometer. The machining is carried out on a conventional lathe. A total of 12 combinations of cutting conditions were set. Specifically, for three cutting speeds of 130, 155 and 180 m·min$^{-1}$, the feed rates of 0.05 and 0.1 mm·rev$^{-1}$ and the cutting widths of 0.2 and 0.35 mm, were evaluated The evaluation assessed the surface quality by both touch and non-touch methods. A structural equation with the appropriate constants and exponents was then constructed from the data obtained using the dynamometer. The experiment confirmed the potential of achieving a value of the average arithmetic profile deviation Ra in the range of 0.3–0.4 when turning hardened steels with cubic boron nitride.

**Keywords:** hardened steel; cubic boron nitride; rolling body; structural equations; surface roughness; cutting force; turning

## 1. Introduction

Despite its difficult machinability, hardened steel is a widely used material for producing the various components of a range of technical equipment, such as the rolling elements of thrust bearings. In their fabrication, hard machining is a frequently used alternative, especially because it is economical. Machining hardened steel often faces limitations in the behavior of the conventional cutting tools used under particular conditions, as reported by Yallese et al. [1]. Therefore, the cutting performance of various cutting tools has been of interest to many researchers in recent years. The ability to predict and evaluate the cutting tool performance enables the required power for machining to be determined and the input energy to be used economically. Osička et al. [2] studied and statistically evaluated the face and flank wear of used tools under different cutting conditions. A strategy for the rapid analysis of tool wear during machining hardened steel was developed by Zahaf et al. [3] based on several experiments under various conditions. Interesting results were produced also by Faris et al. [4] who proved that the concept of using double rake tooling increases the cutting tool performance because of its more favorable stress condition. In their study, Osička et al. [5] extended the already verified cutting conditions of machining by the grinding process. Here, the surface tension state was measured by X-ray diffractometry and the surface quality of the individual samples was statistically evaluated.

Hard steel (45–65 HRC) turning has been a subject of many research studies over the last years. Benga et al. [6] presented a comparison of the performance of four cutting-tool materials when turning bearing steel 100Cr6 hardened to 62–64 HRC with regard to tool life and surface roughness. An interesting experimental investigation was conducted by Thiele et al. [7] to determine the effect of tool edge geometry and workpiece hardness on surface roughness and cutting forces during the hard turning finishing of AISI 52100 steel. Cubic boron nitride inserts with different representative cutting edge modifications were used as cutting tools. This study showed that the effect of cutting edge geometry on surface roughness and cutting forces was statistically significant. It was also found that the two-factor interaction of cutting edge geometry and workpiece hardness had a significant effect on surface roughness. A similar study was conducted by Sahin et al. [8] who presented the surface roughness model with respect to the main cutting parameters such as cutting speed, feed and depth of cut using a response surface methodology. Machining tests were carried out for turning hardened AISI 1050 steels with cubic boron nitride (CBN) cutting tools under different conditions. Based on the experimental data during steel machining, model equations predicting the surface roughness, Ra, Rz and Rmax, were developed.

In investigating the effect of different cutting conditions when machining hardened steel parts, significant attention is often paid to studying the behavior and wear of used cutting tools of various materials. Changes in cutting edge and cutting speed are the frequently monitored parameters that strongly influence the tool wear [9]. Another important parameter is the material of the cutting tool itself. Raczkovi [10] examined the wear of low CBN-content cutting tools for the hard turning of 100Cr6 bearing steel (HRC = 62 ± 2). The performance of the CBN cutting tool, more specifically the (Ti, Al) N-coated CBN tool that is formed on a CBN substrate by the physical vapor deposition method, was tested also by Wada et al. [11]. During their experiments, hardened steel was turned with the (Ti, Al) N-coated CBN tool at cutting speeds of 200 and 300 m·min$^{-1}$, with a feed rate of 0.3 mm·rev$^{-1}$ and a depth of cut of 0.1 mm. In addition, uncoated CBN was also used as a substrate for the (Ti, Al) N coating. Hanel et al. [12] tested an ultra-hard cutting material, specifically nanocrystalline cubic boron nitride (BNNC). This material was fabricated using a high-pressure–high-temperature (HP–HT) process. The starting material was a pyrolytically deposited hexagonal boron nitride (PBN), which was transformed at temperatures of 1400–2200 °C and pressures of 10–20 GPa during direct synthesis without any binder. The average crystallite size of this material was 50–100 nm and it was, therefore, significantly smaller than that of the conventional polycrystalline cubic boron nitride (PCBN) cutting materials. Compared to conventional PCBN cutting materials, this material had an increased hot-hardness and a better temperature resistance. Testing was carried out by turning grooves in hardened steel.

Interesting research in this field was also carried out by Devin et al. [13] who studied the cutting forces and stresses on the surface of cutting tools with inserts made from five sets of cubic boron nitride polycrystals. The inserts had different specific surface areas of the base material. The dynamic strength of the super hard polycrystals was investigated under the diametric compression of disk-shaped specimens with a diameter of 10–12 mm and a thickness of 3–4 mm. An algorithm for the calculation of the probability of failure was described, which included the evaluation of the probability of tool failure by analyzing the differential distribution functions for the compressive and tensile stresses that arose during the turning on the front and back faces of the cutting edge and the distribution of the compressive and tensile strengths of the tool material. This algorithm was used in the development of the VarTool software in the Mathcad package to evaluate the probability of the failure of tools or other structures operating under non-stationary loads.

Overall tool geometry represents another critical condition leading to an increase in cutting efficiency as Harisha et al. [14] point out in their study. Improper tool geometry can result in energy loss and tool wear during the machining of hardened steel, leading to higher production costs. Therefore, the objective of their work was to determine the optimum combination of parameters such as angle of attack, depth of cut, tip radius and

feed rate during turning to reduce the cutting force and energy consumption. Apart from the tool geometry, due attention should also be paid to the cutting temperatures. Pronin et al. [15] address the problem of the cutting temperatures corresponding to the optimum turning modes in a full range of structural alloy hardened steel (HRC 55) used for the manufacture of machine components included in the design of marine vessels. The radial component of the cutting force and the cutting temperature were used as optimization parameters for machining hardened steel with cutting ceramics.

Most of the above-mentioned studies used conventional machining methods while Farahnakian et al. [16] addressed cutting tool wear in machining hard materials and the use of modern machining processes such as ultrasonic assisted turning. In the ultrasonic-assisted turning of hard and brittle materials, the cutting tool failure occurred due to vibration shock. After machining short lengths, microcracks occurred in the tool face, so that sharp cutting edges failed in the initial stages of the machining. Therefore, if the cutting edge is deburred before machining, the cutting edge breakage caused by the vibro-strike condition could be eliminated. The aim of the research was to investigate the flank wear of tungsten carbide tools during the ultrasonically assisted turning of hardened alloy steel and to compare the results with conventional turning.

Surface finish is a very important aspect for designing mechanical elements and is also presented as a quality indicator of the manufacturing processes, which is necessary for proper part geometries [17]. The 3D measurement of technical surfaces plays a crucial role in checking and controlling the properties and the function of materials or engineering parts [17]. Therefore, the effect of different cutting parameters on the surface roughness of machined hardened steel parts is a widely studied phenomenon. Erdem et al. [18] investigated the effect of different cutting parameters on surface roughness Ra and cutting force during hard turning of dry hardened tool steel 1.2367 (55 HRC). In this experimental study, constant depth of cut, three different cutting speeds and three different feeds were selected as the cutting parameters. The effect of the cutting parameters on surface roughness and cutting forces was evaluated by performing an analysis of variance. According to the results, the value of surface roughness increased as the feed rate increased, but the effect of cutting speed on surface roughness was negligible. In contrast to Erdem et al. [18], Aouici et al. [19] considered the depth of cut as a variable when studying the effect of cutting speed, feed rate, workpiece hardness and depth of cut on the surface roughness and cutting force components in hard turning. AISI H11 steel was hardened to (40, 45 and 50) HRC and machined using cubic boron nitride (CBN 7020 from Sandvik), which was essentially composed of 57% CBN and 35% TiCN. Four-factor (cutting speed, feed rate, hardness and depth of cut) and three-level sub-experimental designs were performed, supplemented by a statistical analysis of variance. The results showed that the components of cutting force were significantly affected by depth of cut and workpiece hardness.

The main topic of this article is the discussion of the potential of using experimentally obtained data of individual cutting force components for determining the methodology of mathematical calculation of a particular cutting force component. The theoretical empirical calculation of the structural equation will approximate the actual cutting force value the closer that the actual cutting conditions of the machining are to agreeing with the cutting conditions of the experiment. It must, therefore, be stated here that the derived theoretical calculation of the structural equation of a particular cutting force component will be valid over a certain range of cutting conditions. Thus, it will be possible to predict a relatively simple calculation of the cutting force component without having to make measurements. This is very important in practice, where companies do not have the necessary measuring instruments, but at least need to determine the predicted magnitude of the relevant cutting force components. This knowledge then enables the required power for machining to be determined and the input energy to be used economically.

So far, the applied research has focused mainly on machining functional surfaces of the body and shaft ring [2,5]. This paper's contribution is mainly concerned with the machining of rolling hardened bodies, the real-time measurement of the cutting force

during machining and the determination of the structural equation from the obtained values. These data are not widely available in the literature and have limited validity for specific cutting conditions. Surface roughness parameters after finishing machining are given as a secondary output of this study, considering that the designed cutting conditions must also ensure adequate surface quality. The surface quality in this study was measured by the touch method and for four selected samples by the non-touch method for inter-comparison [20,21].

Despite the long list of similar research studies, this paper provides new information for specific machining conditions and the main result is a structural equation for calculating the cutting force. This calculation option allows a simple way to predict the magnitude of the cutting force without the need for measuring equipment. The coefficients of the equations for calculating the cutting forces are known for standard materials, not for their application-specific hardening treatment. This requirement is based on industrial practice and is linked to the bearing production environment.

## 2. Materials and Methods

The experiment focuses on measuring the cutting forces when machining the functional surface of the component "Rolling Body", which is the outer diameter. The implemented manufacturing process includes:

- Machining hardened components by finishing technology with a CBN tool;
- Carrying out an experiment to measure the cutting forces;
- Evaluation of the force analysis;
- Development of a structural equation;
- Evaluation of the surface quality of the functional surface by the touch method;
- Evaluation of the surface quality of the functional surface by non-contact method.

The experiments were carried out according to the available machinery and laboratory equipment at the Faculty of Mechanical Engineering of Brno University of Technology on the following equipment:

- Universal lathe SV 18 RD;
- Dynamometer Kistler 9257B;
- Taylor Hobson Surtronic S 100 roughness gauge;
- Alicona Infinite Focus G5 non-contact instrument.

The component material of the rolling element was 100 Cr6 (1.3505), according to ISO 683-17. The material composition is in Table 1. The hardness of the samples was 62–64 HRC. The material also complies with the Czech standard ČSN 414109 and the US standard AISI 52100.

**Table 1.** Material composition of 100 Cr6 (%).

| Steel | C | Si | Mn | Cr | Mo | P | S |
|---|---|---|---|---|---|---|---|
| 100Cr6 | 0.93–1.05 | 0.15–0.35 | 0.25–0.45 | 1.35–1.60 | max.0.1 | 0.025 | 0.015 |

### 2.1. Implementation of the Cutting Force Measurement Experiment

The experiment was carried out on 12 samples of rolling elements for larger bearing types.

The approximate dimensions of the sample are shown in Figure 1:

- Outer diameter (mm)　　　　32;
- Length (mm)　　　　　　　　60.

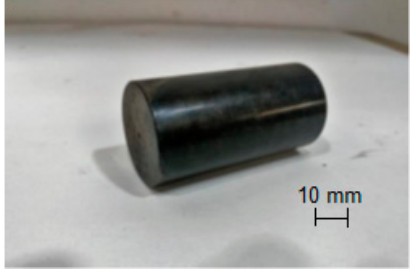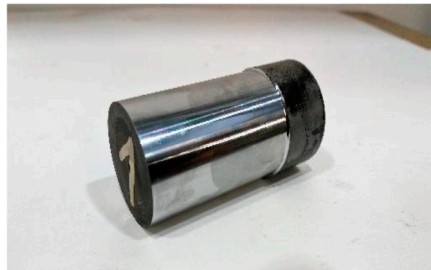

**Figure 1.** Sample of a semi-finished rolling element, left, and a machined sample, right.

Unlike previous experiments with bearing rings, the SP280 SY machine was not used because of the near-impossible placement of the Kistler measuring system on this machine, namely the placement of the measuring probes on the vertical slide of the machine. In previous experiments [2] cutting tools from SECO were used, but in this experiment alternative tools from Dormer Pramet were applied.

### 2.2. Technological Conditions of Machining When Measuring with a Kistler Dynamometer

The machine used for the experiment was the universal spindle lathe SV 18 RD, which is a conventional lathe that has been verified for rigidity. Verification of the machine's accuracy and rigidity was carried out using a certified method as part of preventive maintenance. The machine also has a considerable range of cutting speeds that can be continuously controlled. The spindle speed range is 56–2800 min$^{-1}$. The power of the main motor at maximum rpm is 10 kW. The main advantage of this conventional lathe is the added potential for placing the measuring probes of the Kistler dynamometer on the back of the slide.

The specific tools used by Dormer Pramet were as follows:

- Tool holder PCLNL 2525 M 12 in left-hand version;
- Replaceable insert CNGA 120408 S 01020B shown in Figure 2, made of TB310, polycrystalline cubic boron nitride, suitable for use without cutting fluid.

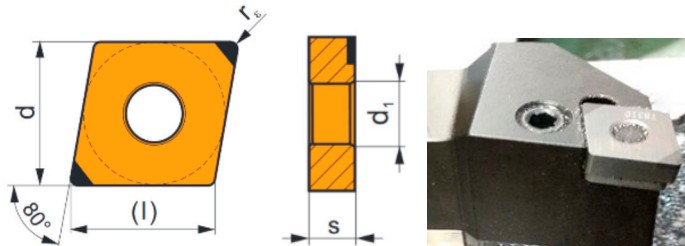

**Figure 2.** General dimensions of the insert and toolholder.

One tool was used for the entire series of 12 samples. The insert dimensions and recommended cutting conditions are given in Table 2.

**Table 2.** Dimensional specifications of the insert used and range of recommended cutting conditions.

| $r_\varepsilon$ | $a_p$ | $f$ | $v_c$ | $d$ | $d_1$ | $l$ | $s$ |
|---|---|---|---|---|---|---|---|
| (mm) | (mm) | (mm·rev$^{-1}$) | (m·min$^{-1}$) | (mm) | (mm) | (mm) | (mm) |
| 0.8 | 0.1–2.7 | 0.02–0.20 | 100–200 | 12.7 | 5.16 | 12.9 | 4.76 |

The cutting conditions are represented the following variables:

- Cutting speed $v_c$, which depends on the number of revolutions and diameter of the workpiece;
- The feed rate $f$, which is defined by the movement of the tool per revolution;

- The cutting edge width $a_p$, which is determined by the tool's approach to the cutting edge.

The geometry of cutting tools is an important factor in turning hard materials. The basic geometry of a cubic boron nitride insert as laid on a horizontal surface as follows:

- Orthogonal face angle $\gamma_o = 0°$;
- Orthogonal back angle $\alpha_o = 0°$;
- Orthogonal edge angle $\beta_0 = 90°$.

After inserting the insert into the tilted toolholder bed, the working angles of the insert are as follows:

- Orthogonal face angle $\gamma_o = -6°$;
- Orthogonal back angle $\alpha_o = 6°$;
- Blade inclination angle $\lambda_s = -6°$;
- Main blade setting angle $\kappa_r = 95°$;
- Angle of adjustment of the secondary blade $\kappa_r' = 5°$.

The ranges of $f$ and $a_p$ given here are informative; the parameters used in the study provided data to determine the structural equation. One tool was used for the whole series of samples.

The complete tool holder, including the inserts used during the experiment, as placed on the probe of the Kistler 9257B dynamometer is shown in Figure 3a. The diagram of the distribution of cutting forces during turning is given in the schematic part of Figure 3b and shows the normal situation when the tool is in front of the axis. In our case, the tool was behind the axis, as shown in Figure 3a, with the actual direction of the forces and a description of the situation given in this Figure. This location is required by the design of the Kistler dynamometer encoder. The direction of the passive force $F_p$ is then reversed. In the last part of Figure 3c, the label of the Kistler 9257B dynamometer is shown with the individual directions of the axes of the cutting resistance against the cutting forces when the dynamometer is placed in the front part of the caliper. This then corresponds to the directions of the x-axis for the sliding force $F_f$, the z-axis for the cutting force $F_c$ and the y-axis for the passive force $F_p$.

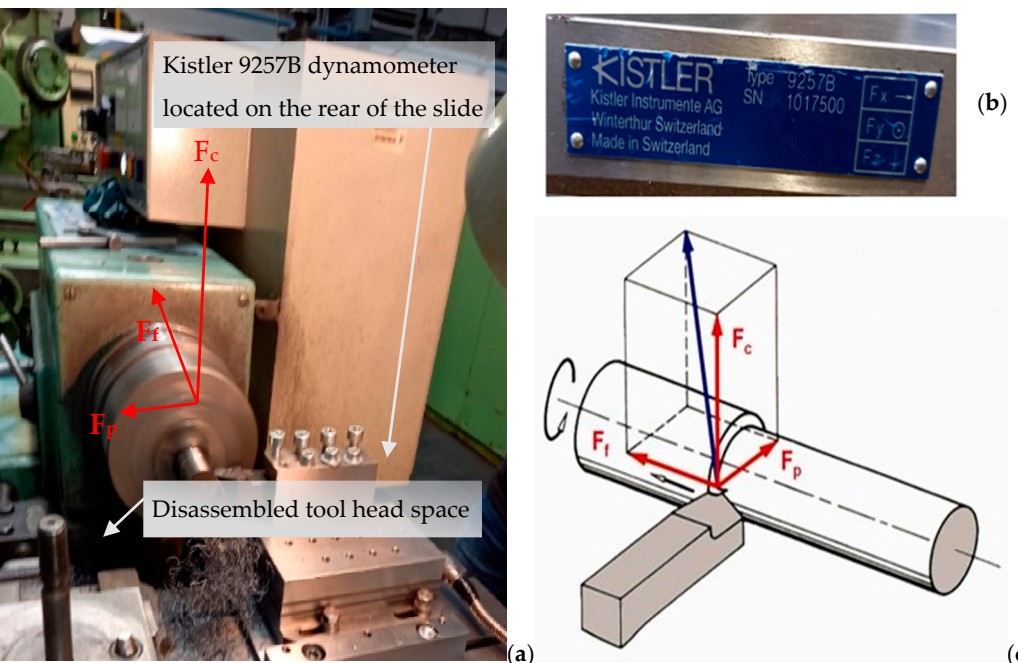

**Figure 3.** Diagram of the location of the Kistler 9257B dynamometer during the experiment (**a**), table indicating the directions of cutting resistance (**c**) and diagram of the distribution of cutting forces during turning (**b**) [22].

The cutting conditions for machining were chosen according to the experience from previous experiments [2], but were applied as a wider range, so that the mathematical dependencies could be subsequently determined. The experiment was planned within the DOE methods using Minitab software. The main factors chosen were $v_c$, $f$ and $a_p$. A full factorial design of the experimental plan was chosen. The specific values of the cutting conditions are in Table 3. These are always four combinations of feed rate, $f$, and blade cutting width, $a_p$, at three different cutting speeds, $v_c$. This range of cutting conditions was chosen to cover the possible cutting conditions for finishing machining. At the same time, this range of cutting conditions allows empirical data to be obtained for the construction of the structural equation.

**Table 3.** The cutting conditions of the experiment, showing the full factorial plan.

| Sample No. | $v_c$ (m·min$^{-1}$) | F (mm·rev$^{-1}$) | $a_p$ (mm) |
|---|---|---|---|
| 1 | | 0.05 | 0.2 |
| 2 | | 0.05 | 0.35 |
| 3 | 130 | 0.1 | 0.2 |
| 4 | | 0.1 | 0.35 |
| 5 | | 0.05 | 0.2 |
| 6 | | 0.05 | 0.35 |
| 7 | 155 | 0.1 | 0.2 |
| 8 | | 0.1 | 0.35 |
| 9 | | 0.05 | 0.2 |
| 10 | | 0.05 | 0.35 |
| 11 | 180 | 0.1 | 0.2 |
| 12 | | 0.1 | 0.35 |

The machining was carried out without the processing fluid, and due to the geometry of the cutting insert (without a chip sealer), a continuous segmented chip was realized, as shown in Figure 4. This chip breaks easily and does not damage the already machined surface. A detail of the chip is shown in Figure 4. This chip character is due to the turbid state of the stock material and was achieved with all combinations of the cutting parameters.

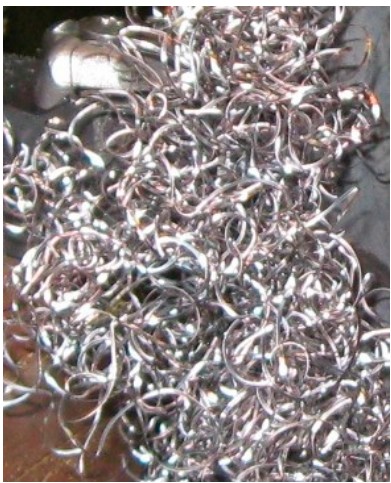

**Figure 4.** Detail of a continuous chip divided into sub-segments.

### 2.3. Measurement of Cutting Forces with a Kistler Dynamometer

A Kistler type 9257B, SN1017500 stationary dynamometer was used to evaluate the cutting force measurements. During longitudinal turning, the force components were measured in three directions according to the coordinate system, i.e., in the x, y and z axes,

corresponding to the force components, specifically the cutting force, $F_c$, the passive force, $F_p$, and the feed force, $F_f$. The measuring apparatus consisted of a dynamometer, a Kistler 5070A hub amplifier and a data acquisition and analysis system by means of which the data were transferred to a computer. As each sub-sample was machined, a measurement was run and after turning was completed, the values were recorded and stored in DynoWare. The measurement was set to 60 s to cover all the machine times when machining each sample. The sampling rate was set to 2000 Hz. The unwanted extreme measurement dates were filtered out.

### 2.4. Surface Quality Measurement by the Touch Method

Measurements of individual surface roughness parameters were made with a Taylor Hobson Surtronic S-128 roughness tester. The length of the elementary section was 4 mm and 3 measurements were made on each sample. The advantage of touch measurement lies in the simplicity of operating the measuring instrument with the potential of an easy application in the workshop operation. However, the touch method does not allow the display of area parameters. In terms of investment costs, it is considerably cheaper than the non-touch method.

### 2.5. Surface Quality Measurement by Non-Contact Method

The evaluation of the surface quality of the functional surface by the non-contact method was carried out on the Alicona Infinite Focus G5 (hereafter referred to as Alicona). It is a highly accurate, fast and flexible optical 3D measurement system. The 3D measurements were made with Focus Variation. Focus Variation combines the small depth of focus of an optical system with vertical scanning to provide topographical and color information from the variation of focus [23]. The main problem of optical instruments is that most existing roughness standards are relatively smooth and can hardly be measured with several optical instruments [24]. The main component of the system is precision optics containing various lens systems that can be equipped with different objectives, allowing measurements with different resolutions [23]. With a beam splitting mirror, light emerging from a white light source is inserted into the optical path of the system and focused onto the specimen via the objective [24]. Depending on the topography of the specimen, the light is reflected into several directions as soon as it hits the specimen via the objective [23]. If the topography shows diffuse reflective properties, the light is reflected equally strongly in each direction [23]. In the case of specular reflections, the light is scattered mainly in one direction [23]. The main disadvantage is with the reflectivity of some surfaces, which means that more lenses and measuring conditions are required [7,8]. The Alicona is a highly accurate, extremely fast and flexible optical 3D micro-coordinate measurement system for both form and roughness measurement, with a vertical resolution of down to 10 nm [21]. This measuring device is ideal for most precise surface analyses on homogeneous as well as mixed materials, and the system produces exact topographic information in true colors using vertical scanning and a shallow depth of field [21]. The Laboratory Measurement Module 6.6.1 was used for measuring the surface roughness parameters [25].

The measurements were carried out under the following conditions:

- Using a $50\times$ lens;
- Filter was Lc [μm] 800;
- Profile measurement path length [mm] was 4.

### 3. Results

The results include the issue of cutting forces and the evaluation of surface integrity by both touch and non-touch methods. Due to the large amount of data, the graphical representation is limited to selected samples of components, namely parts 5, 6, 8 and 12.

### 3.1. Evaluation of Cutting Force Measurements with the Kistler Dynamometer

The measurement data were converted to a .txt file and then graphical dependencies of the time course on the force load were created in Microsoft Excel. Due to the very small scatter of individual force values in virtually all graphs, these are sufficient as input data for the structural equation. The average values of the individual cutting forces at the specified cutting conditions are in Table 4.

**Table 4.** Average values of individual cutting forces.

| Sample No. | $v_c$ (m·min$^{-1}$) | $f$ (mm·rev$^{-1}$) | $a_p$ (mm) | Force $F_c$ (N) | Force $F_f$ (N) | Force $F_p$ (N) | Resulting Force $F$ (N) |
|---|---|---|---|---|---|---|---|
| 1 | | 0.05 | 0.2 | 121.4 | 49.0 | 71.0 | 148.9 |
| 2 | | 0.05 | 0.35 | 165.6 | 99.7 | 114.1 | 224.5 |
| 3 | 130 | 0.1 | 0.2 | 167.9 | 64.8 | 112.8 | 212.4 |
| 4 | | 0.1 | 0.35 | 224.9 | 117.4 | 175.5 | 308.5 |
| 5 | | 0.05 | 0.2 | 133.6 | 54.8 | 72.3 | 161.5 |
| 6 | | 0.05 | 0.35 | 153.9 | 106.6 | 115.3 | 219.9 |
| 7 | 155 | 0.1 | 0.2 | 171.7 | 59.5 | 97.3 | 206.2 |
| 8 | | 0.1 | 0.35 | 206.1 | 128.3 | 175.5 | 299.6 |
| 9 | | 0.05 | 0.2 | 119.0 | 47.1 | 68.7 | 145.3 |
| 10 | | 0.05 | 0.35 | 147.5 | 87.7 | 103.3 | 199.0 |
| 11 | 180 | 0.1 | 0.2 | 152.1 | 62.2 | 123.2 | 205.5 |
| 12 | | 0.1 | 0.35 | 199.1 | 118.6 | 197.4 | 304.5 |

Figures 5–8 show the force load time histories for the selected samples, including the calculated total cutting force $F$. For all machined samples, one pass and one measurement was performed on each sample.

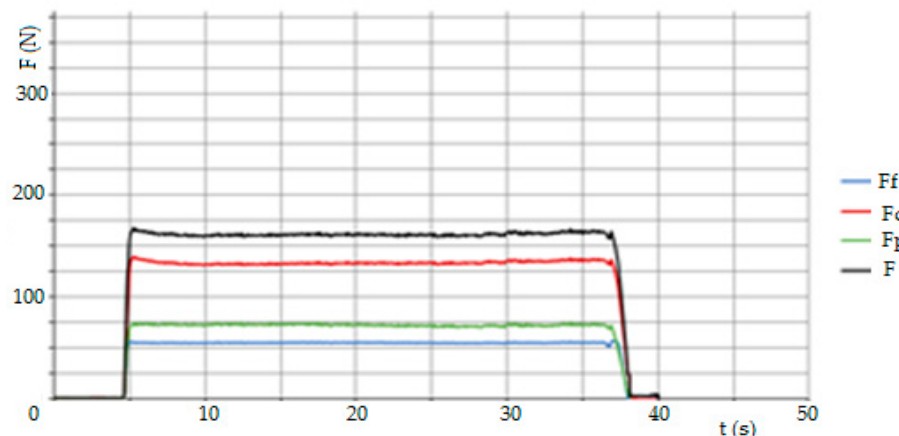

**Figure 5.** Force load history of sample 5 ($v_c$ = 155 m·min$^{-1}$, $f$ = 0.05 mm·rev$^{-1}$, $a_p$ = 0.2 mm).

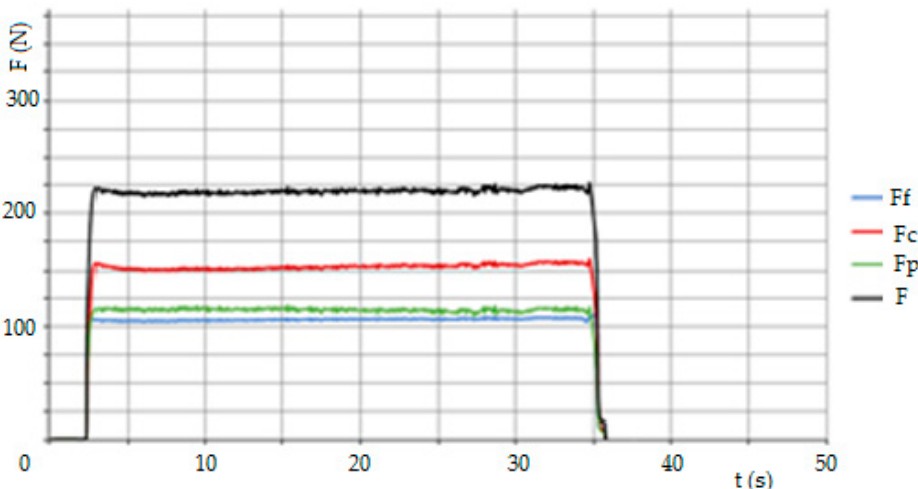

**Figure 6.** Force load history of sample 6 ($v_c$ = 155 m·min$^{-1}$, $f$ = 0.05 mm·rev$^{-1}$, $a_p$ = 0.35 mm).

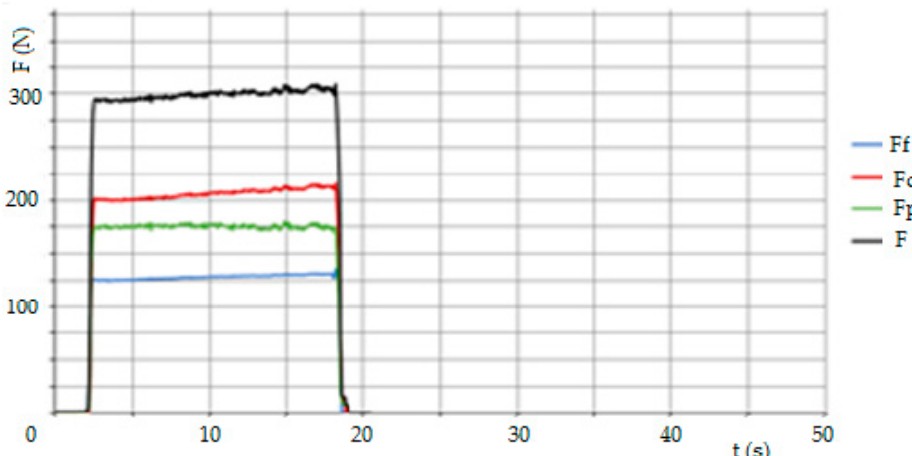

**Figure 7.** Force load history of sample 8 ($v_c$ = 155 m·min$^{-1}$, $f$ = 0.1 mm·rev$^{-1}$, $a_p$ = 0.35 mm).

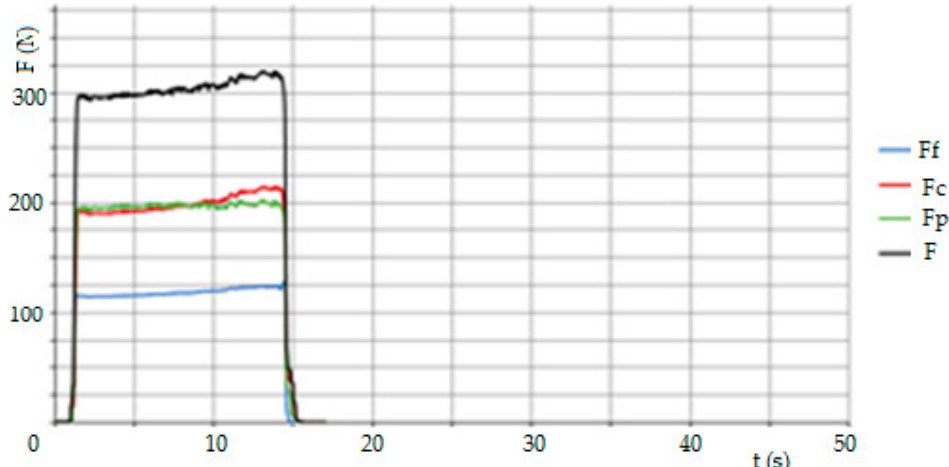

**Figure 8.** Force load history of sample 12 ($v_c$ = 180 m·min$^{-1}$, $f$ = 0.1 mm·rev$^{-1}$, $a_p$ = 0.35 mm).

In order to determine the effect of the cutting parameters, the data obtained from the experimental measurements were analyzed by General Factorial Regression analysis using the DOE plan and Minitab software.

A basic analysis provided information on the combinations of factors and their effects and interactions. The results in Table 5 show that due to the low $p$ value of 0.003, $a_p$ and $f$ are the main factors. Considering that the $p$-Value = 0.003, it can be concluded that these factors are significant with a reliability of 99.7%. Cutting speed, $v_c$, is almost negligible in terms of the effect of the $F_c$ and their interactions with $a_p$. The interaction with $f$ is moderate in terms of the significance level.

**Table 5.** General Factorial Regression.

| Source | DF | Adj. SS | Adj MS | F-Value | *p*-Value |
|---|---|---|---|---|---|
| Model | 9 | 20,301.8 | 2255.75 | 81.69 | 0.012 |
| Linear | 4 | 18,992.1 | 4748.02 | 171.95 | 0.006 |
| $v_c$ | 2 | 131.2 | 65.60 | 2.38 | 0.296 |
| $f$ | 1 | 9464.1 | 9464.08 | 342.74 | 0.003 |
| $a_p$ | 1 | 9396.8 | 9396.80 | 340.30 | 0.003 |
| 2-Way | 5 | 1309.7 | 261.93 | 9.49 | 0.098 |
| Interactions | | | | | |
| $v_c{*}f$ | 2 | 533.7 | 266.86 | 9.66 | 0.094 |
| $v_c{*}a_p$ | 2 | 33.3 | 16.66 | 0.60 | 0.624 |
| $f{*}a_p$ | 1 | 742.6 | 742.61 | 26.89 | 0.035 |
| Error | 2 | 55.2 | 27.61 | | |
| Total | 11 | | | | |

The methodology for determining the specific form of the structural equation requires the following simplification. As shown in Table 5, the influence of the cutting speed is minimal, which implies the assumption of a very small value of the exponent $z_{Fc}$ of about $\pm0.05$. Graphically, the interactions and dependencies are depicted in Figure 9. The low influence of $v_c$ and the high influence of $f$ and $a_p$, due to the orthogonal distribution, are clearly visible.

**Figure 9.** Main effect plot of individual variables on the cutting force $F_c$.

Figure 10 shows a contour plot of the effect of $a_p$ and $f$ on the cutting force $F_c$. By referring to the contour representation, it is possible to observe the relationships and optimize the process, also the uniform distribution of the dependence for both $f$ and $a_p$ can be seen.

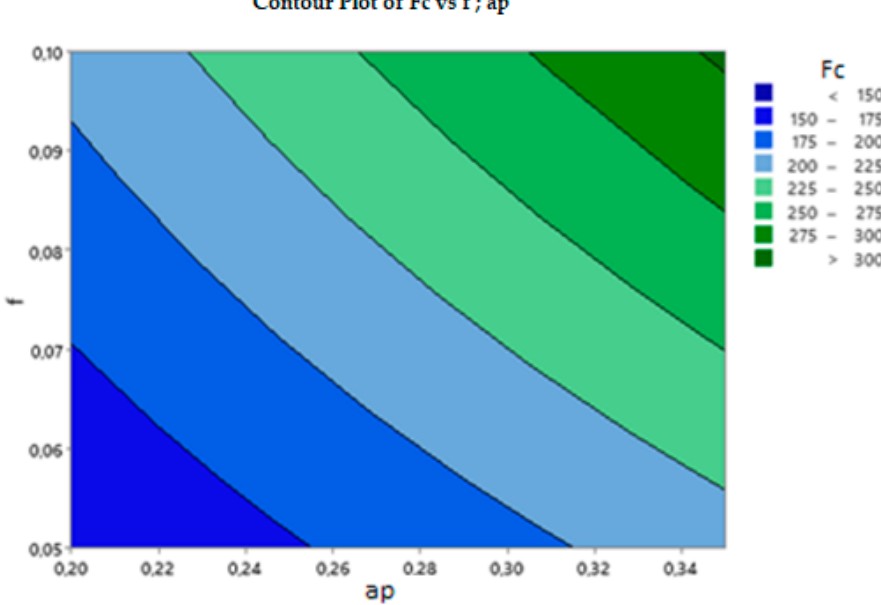

**Figure 10.** Contour plot of the effect of $f$ and $a_p$.

*3.2. Derivation of the Structural Equation*

The derivation of the structural equation for the cutting force $F_c$ for turning material in the hardened state is based on the data in Table 4 and on the following Equation (1):

$$F_C = C_{F_C} \cdot a_p^{x_{F_C}} \cdot f^{y_{F_C}} \cdot v_c^{z_{F_C}} \tag{1}$$

This relationship is empirical and depends on the constant $C_{Fc}$, the exponents $x_{Fc}$, $y_{Fc}$ and $z_{Fc}$ and the variables—in this case the blade width, $a_p$, the feed, $f$, and possibly the cutting speed, $v_c$. The values of the constant $C_{Fc}$ and the exponents $x_{Fc}$, $y_{Fc}$ and $z_{Fc}$, depend on the specific machining conditions and are valid within a certain range. Furthermore, the type of material and its condition have an influence, which can be expressed by the machinability class. When calculating the cutting force according to this equation, we have to consider an inaccuracy which will be proportional to the difference between our machining conditions and those used to calculate the constant $C_{Fc}$ and the exponents $x_{Fc}$, $y_{Fc}$ and $z_{Fc}$. The cutting speed term in the general formula is therefore not included and its possible influence is reflected in the value of the constant $C_{Fc}$. From Table 4 the individual values of $Fc$ are averaged for the combinations of feed $f$ and cutting edge width $a_p$ used and listed in Table 6 for use in the following calculation:

$$F_C = C_{F_C} \cdot a_p^{x_{F_C}} \cdot f^{y_{F_C}} \implies C_{F_C} = \frac{F_C}{a_p^{x_{F_C}} \cdot f^{y_{F_C}}} \tag{2}$$

**Table 6.** Averaged values of cutting force $F_c$ for combinations of feed $f$ and cutting edge width $a_p$.

| Combination $f$ and $a_p$ | $f$ (mm·rev$^{-1}$) | $a_p$ (mm) | $F_c$ (N) |
|:---:|:---:|:---:|:---:|
| 1 | 0.05 | 0.2 | 124.7 |
| 2 | 0.05 | 0.35 | 155.6 |
| 3 | 0.1 | 0.2 | 163.9 |
| 4 | 0.1 | 0.35 | 210.3 |

Equation (2) in its modified form is the basic formula for determining the methodology for determining the values of the exponents $x_{Fc}$, $y_{Fc}$ and subsequently the constant $C_{Fc}$.

### 3.2.1. Derivation of the Exponent $x_{Fc}$

To derive the exponent $x_{Fc}$, a constant displacement value $f$ is assumed and the equation takes the following form (3):

$$F_C = C_{F_{a_p}} \cdot a_p^{x_{FC}} \tag{3}$$

After logarithmization, $x_{Fc}$ can be defined as the directive of the line *tg α*:

$$\log F_C = \log C_{F_{a_p}} + x_{F_C} \log a_p \tag{4}$$

A graphical representation is shown in Figure 11.

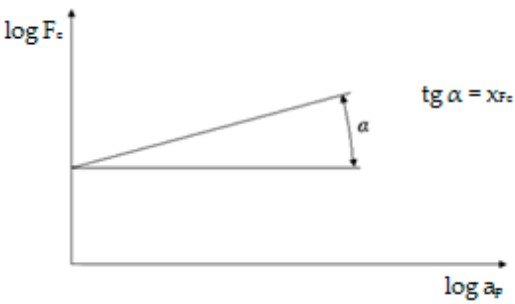

**Figure 11.** Graphical representation of $x_{Fc}$.

Equation (5) to calculate $x_{Fc}$ from the data in Table 6:

$$x_{F_C} = tg\alpha = \frac{\log F_{C2} - \log F_{C1}}{\log a_{p2} - \log a_{p1}} \tag{5}$$

After setting the values of the combination of cutting conditions for a constant feed $f$ 0.1 mm:

$$x_{F_C} = tg\alpha = \frac{\log 210.3 - \log 163.9}{\log 0.35 - \log 0.2} = 0.444 \tag{6}$$

### 3.2.2. Derivation of the Exponent $y_{Fc}$

To derive the exponent $y_{Fc}$, a constant value of the cutting edge width $a_p$ is assumed and the equation takes the following form (7):

$$F_C = C_{F_f} \cdot f^{y_{FC}} \tag{7}$$

After logarithmization, $y_{Fc}$ can be defined as the directive of the line *tg α*:

$$\log F_C = \log C_{F_f} + y_{F_C} \log f \tag{8}$$

A graphical representation is shown in Figure 12.

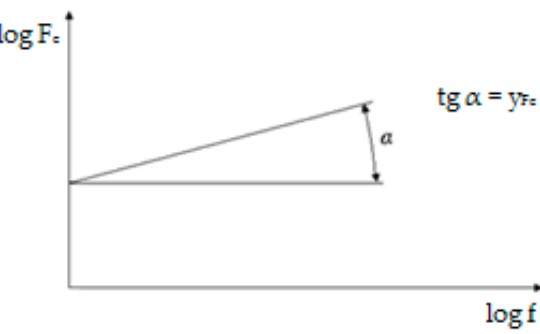

**Figure 12.** Graphical representation of $y_{Fc}$.

Equation (9) to calculate $y_{Fc}$ from the data in Table 6:

$$y_{F_C} = tg\alpha = \frac{\log F_{C2} - \log F_{C1}}{\log f_2 - \log a_1} \tag{9}$$

After setting the values of the combination of cutting conditions for a constant cutting edge width where $a_p$ is 0.35 mm:

$$y_{F_C} = tg\alpha = \frac{\log 210.3 - \log 155.6}{\log 0.1 - \log 0.05} = 0.437 \tag{10}$$

### 3.2.3. Determination of the Constant $C_{Fc}$

Given the already calculated values of the exponents $x_{Fc}$ and $y_{Fc}$, the constant $C_{Fc}$ can be determined using Equation (2) by substituting one of the combinations of the blade width $a_p$, the feed rate $f$ and the corresponding averaged value of the measured cutting force $F_c$ from Table 6:

$$C_{F_C} = \frac{210.3}{0.35^{0.444} \cdot 0.1^{0.437}} = 918.34 \tag{11}$$

### 3.2.4. Final Form of the Structural Equation

The final form of the structural equation for the calculation of $F_c$ after adding the rounded numerical values of the constant $C_{Fc}$, and the exponents $x_{Fc}$, $y_{Fc}$, is given in Equation (12):

$$F_C = 918 \cdot a_p^{0.44} \cdot f^{0.44} [N] \tag{12}$$

Given virtually the same value of the exponents $x_{Fc}$, $y_{Fc}$, the structural equation can be written in the simplified form (13):

$$F_C = 918 \cdot (a_p \cdot f)^{0.44} [N] \tag{13}$$

This form of the structural equation holds for the range of cutting conditions corresponding to the experiment. This range can be extended in the following way:

- Material 10 Cr6 in hardened condition (HRC)    62–64;
- Cutting edge width $a_p$ (mm)    0.1–0.45;
- Feed $f$ (mm·rev$^{-1}$)    0.05–0.15;
- Cutting speed $v_c$ (m·min$^{-1}$)    110–200.

### 3.3. Evaluation of the Surface Quality of the Functional Area

The evaluation of the quality of the functional area was carried out preferentially by the touch method and for the verification of the measured data the measurement was carried out by the non-contact method.

### 3.3.1. Evaluation of the Surface Quality of the Functional Area by the Touch Method

The measurements of individual surface roughness parameters were performed with a Taylor Hobson Surtronic S-128 roughness tester. Each measurement was performed three times and Table 7 shows the averaged values of selected surface quality parameters according to ISO 4287.

**Table 7.** Values of selected surface quality parameters obtained under the specified cutting conditions.

| Sample No. | $v_c$ (m·min$^{-1}$) | $f$ (mm·rev$^{-1}$) | $a_p$ (mm) | Ra (μm) | Rz (μm) | Rmr (%) |
|---|---|---|---|---|---|---|
| 1 | | 0.05 | 0.2 | 0.38 | 2.27 | 48.9 |
| 2 | 130 | 0.05 | 0.35 | 0.34 | 2.17 | 48.7 |
| 3 | | 0.1 | 0.2 | 0.36 | 2.17 | 46.7 |
| 4 | | 0.1 | 0.35 | 0.37 | 2.23 | 47.7 |
| 5 | | 0.05 | 0.2 | 0.27 | 1.63 | 52.6 |
| 6 | 155 | 0.05 | 0.35 | 0.23 | 1.43 | 49.4 |
| 7 | | 0.1 | 0.2 | 0.35 | 2.17 | 48.8 |
| 8 | | 0.1 | 0.35 | 0.48 | 2.87 | 47.9 |
| 9 | | 0.05 | 0.2 | 0.28 | 1.73 | 51.1 |
| 10 | 180 | 0.05 | 0.35 | 0.36 | 2.07 | 50 |
| 11 | | 0.1 | 0.2 | 0.3 | 1.9 | 50.7 |
| 12 | | 0.1 | 0.35 | 0.3 | 1.93 | 50.4 |

The data marked in green were selected for measurement by the non-contact method for the following reasons. Samples 5 and 6 have the best surface quality result measured by the touch method. In contrast, sample 8 has the worst surface quality result. Sample 12 was selected because of the maximum cutting conditions used. The other samples were not measured by non-contact for capacity reasons. For the surface roughness parameter, the standard deviation values of the average arithmetic deviation of the Ra profile are given in Table 8 below.

**Table 8.** Standard deviation values of the average arithmetic deviation of the Ra profile.

| Sample No. | Ra Measured (μm) | | | Ra Average (μm) | Standard Deviation |
|---|---|---|---|---|---|
| 1 | 0.41 | 0.40 | 0.32 | 0.38 | 0.049 |
| 2 | 0.33 | 0.39 | 0.30 | 0.34 | 0.046 |
| 3 | 0.34 | 0.33 | 0.41 | 0.36 | 0.044 |
| 4 | 0.38 | 0.37 | 0.37 | 0.37 | 0.006 |
| 5 | 0.27 | 0.27 | 0.28 | 0.27 | 0.006 |
| 6 | 0.21 | 0.27 | 0.20 | 0.23 | 0.038 |
| 7 | 0.38 | 0.29 | 0.37 | 0.35 | 0.049 |
| 8 | 0.47 | 0.49 | 0.49 | 0.48 | 0.012 |
| 9 | 0.26 | 0.28 | 0.30 | 0.28 | 0.020 |
| 10 | 0.41 | 0.38 | 0.28 | 0.36 | 0.068 |
| 11 | 0.31 | 0.29 | 0.29 | 0.30 | 0.012 |
| 12 | 0.28 | 0.31 | 0.31 | 0.30 | 0.017 |

The best result in terms of surface quality is for samples 5 and 6 at cutting conditions that are proven to work on larger diameters of bearing rings.

### 3.3.2. Evaluation of the Surface Quality of the Functional Area by the Non-Contact Method

The evaluation of the surface quality of the functional surface by the non-contact method was carried out using the Alicona instrument for samples 5 and 6, where the best results were obtained, and for sample 8, where the worst results were obtained. For relevance, sample 12 with the highest values of cutting conditions was also assessed. The results are shown in Table 9.

**Table 9.** Values of selected surface quality parameters measured on Alicona.

| Sample No. | Ra (μm) | Rz (μm) | Rk (μm) | Rpk (μm) | Rvk (μm) | Rmr1 (%) | Rmr2 (%) |
|---|---|---|---|---|---|---|---|
| 5 | 0.273 | 1.459 | 0.883 | 0.217 | 0.255 | 11.77 | 91.83 |
| 6 | 0.232 | 1.471 | 0.547 | 0.296 | 0.35 | 7.59 | 74.62 |
| 8 | 0.328 | 1.644 | 1.068 | 0.191 | 0.175 | 10.65 | 94.85 |
| 12 | 0.196 | 1.302 | 0.665 | 0.269 | 0.186 | 9 | 94.36 |

The meanings of the abbreviations are as follows, according to ISO4287:

Ra—Average arithmetic deviation of the profile;
Rz—Maximum profile height;
Rmr—Mutual material ratio for (Mr = 50%, an offset Rδc = 0.1 μm);

and according to ISO13565:

Rk—Core roughness depth, height of the core material;
Rpk—Reduced peak height, mean height of the peaks above the core material;
Rvk—Reduced valley height, mean depth of the valleys below the core material;
Rmr1—Peak material component, the proportion of peaks above the core material;
Rmr2—Peak material component, the fraction of the surface which will carry the load.

The comparison of Ra and Rz values in the touch and non-contact methods shows almost the same values for samples 5 and 6, while better values were measured in samples 8 and 12 by the non-contact method. In the following Figures 13–18 detailed representations of the progression of the individual variables for the samples are given.

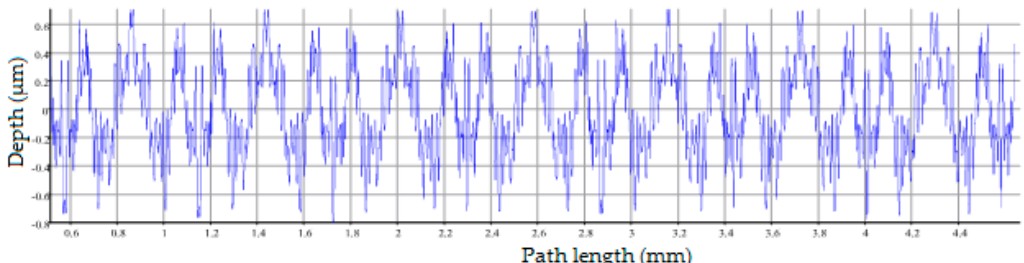

**Figure 13.** Profile measurement part 5: Ra 0.273 μm Rz 1.459 μm.

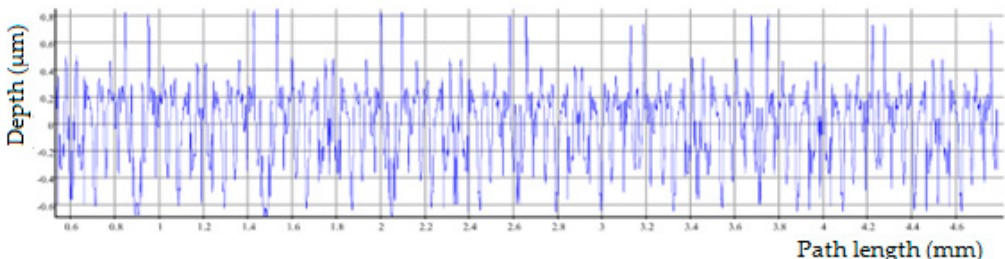

**Figure 14.** Profile measurement part 6: Ra 0.232 μm Rz 1.471 μm.

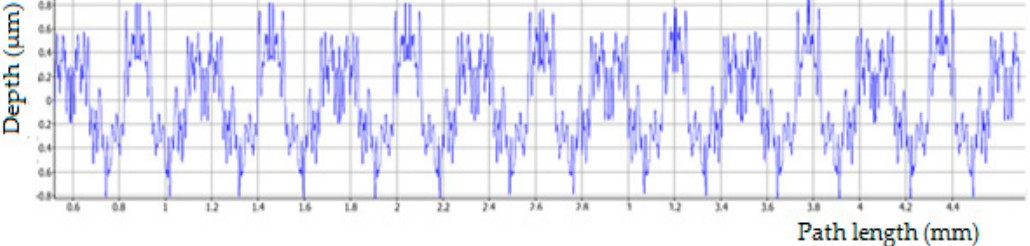

**Figure 15.** Profile measurement part 8: Ra 0.328 μm, Rz 1.644 μm.

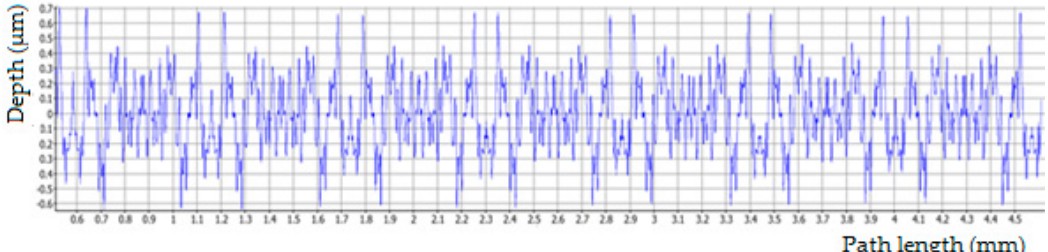

**Figure 16.** Profile measurement part 12: Ra 0.196 µm, Rz 1.302 µm.

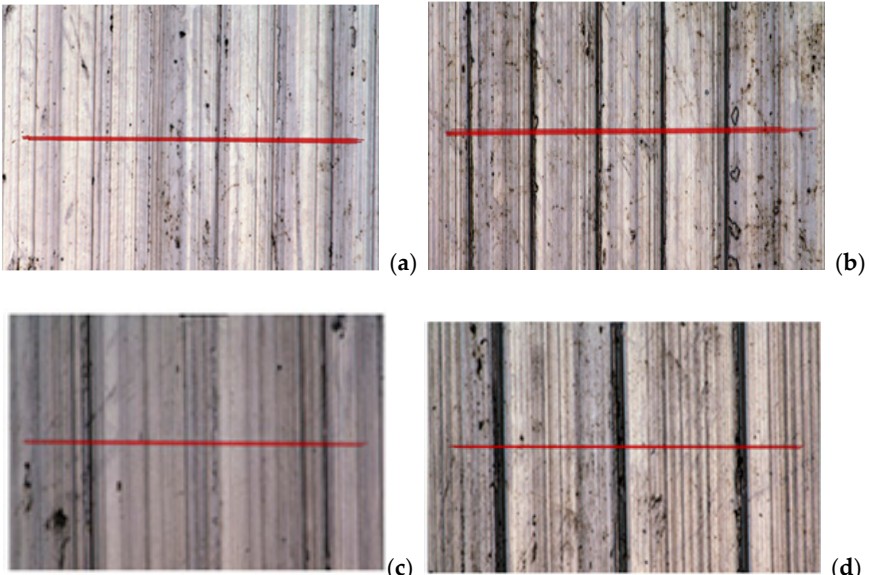

**Figure 17.** Profile measurement display of the surface: (**a**) part 5, (**b**) part 6, (**c**) part 8, (**d**) part 12.

Figures 13–16 show the surface roughness information from the line section shown for the samples in Figure 17. From the line sections, the highest roughness value for part 8 can be seen, there is a cyclic pattern which is determined by the magnitude of the displacement *f*. For part 12 the lowest roughness was measured, and a repeating pattern can be observed here as well, but the size of the individual paths is not as apparent and other influences, not just the magnitude of the displacement, are evident.

The most regular profile roughness is shown in both Figures 14 and 16.

Figure 18 shows a 3D surface with false color visualization for Rz from which the planar waveform is well observed, showing that the linear evaluation from Figures 13–16 is indicative of the planar waveform of the sample roughness. The best result in the spatial contrast representation is given by part 12 in Figure 18d.

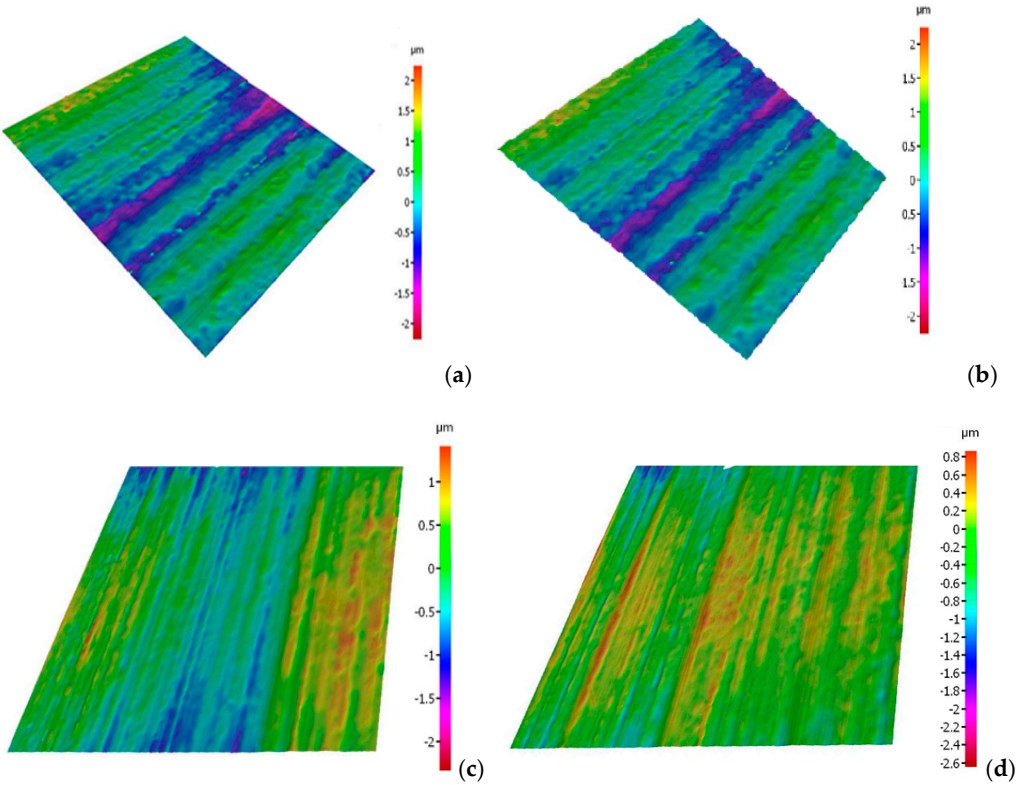

**Figure 18.** Spatial contrast profile display of the surface: (**a**) part 5, (**b**) part 6, (**c**) part 8, (**d**) part 12.

## 4. Discussion

The input data for the evaluation of surface roughness in all samples were mainly the measurements by the touch method, where three measurements were made for each sample. This number allows the determination of at least the size of the standard deviation of the measurements for each sample. However, the set of three measurements is too small for the calculation of other statistical variables. The non-contact method was used for four samples to compare the results, and here one measurement was taken for each sample. The experiment conducted was primarily focused on the measurement of the cutting force components on a conventional machine tool. The surface quality data is a secondary output of the experiment and is intended to confirm the feasibility of using a CBN tool. A detailed evaluation of the results on individual samples is as follows.

When the cutting speed is set to $v_c$ = 130 m·min$^{-1}$, there are no significant differences due to changes in the cutting width $a_p$ and feed per revolution $f$. In general, the resulting surface finish depends mainly on the feed rate $f$; the lower the feed rate f, the lower the surface finish. The average values of the arithmetic deviation of the profile under consideration, Ra, are between 0.3 and 0.4 μm. For a cutting speed of $v_c$ = 130 m·min$^{-1}$, the best combination of cutting conditions appears to be $a_p$ = 0.35 mm and $f$ = 0.05 mm·rev$^{-1}$, with an average Ra value of 0.34 μm. The best (i.e., the smallest) values of the Ra parameter were obtained for a cutting speed of $v_c$ = 155 m·min$^{-1}$ in combination with $a_p$ = 0.35 mm and $f$ = 0.05 mm·rev$^{-1}$, where the average Ra value is 0.23 μm, i.e., sample 6. However, for the same cutting speed $v_c$ and increasing the feed per revolution to $f$ = 0.1 mm·rev$^{-1}$ with the same $a_p$ value, the average Ra value was 0.48 μm, an increase of approximately 209% (for sample 8). These fluctuations may also be due to inaccuracies during the measurement or a crack may appear on the surface of the sample that was measured. These factors can be eliminated by repeating the measurements more often. At a cutting speed of $v_c$ = 180 m·min$^{-1}$ it can be observed that more stable surface quality values are recorded in connection with the increase of the feed per revolution $f$ from 0.05 mm to 0.1 mm. The average value of the parameter Ra is at 0.3 μm. Again, the small number of measurements

is evident here, specifically for sample 10 ($f$ = 0.05 mm·rev$^{-1}$ and $a_p$ = 0.35 mm), where the resulting Ra values fluctuate more, as seen in the highest value of the standard deviation. As for the resulting average Ra values, for samples 5, 6, 9, 11 and 12, a value lower than Ra = 0.30 μm was obtained. These samples have cutting speeds of $v_c$ = 155 m·min$^{-1}$ and $v_c$ = 180 m·min$^{-1}$. In general, based on the results obtained from the contact measurements, it can be stated that out of 12 machined samples with different combinations of cutting conditions, 11 samples were below Ra = 0.4 μm on average during the measurements. It is difficult to reach this value with conventional finishing tools made of materials other than CNB. It follows that the CNB insert material is able to replace finishing operations such as grinding in certain cases. After performing comparative measurements of selected samples on the Alicona instrument, it can be stated that for samples 5 and 6 almost the same values were obtained as for the touch method. For sample 8, where the touch method measured an average value of Ra = 0.48, the non-contact method measured Ra = 0.328, which also places this sample among the compliant pieces. For sample 12, where the average value of Ra = 0.3 was measured by the touch method, the average value of Ra = 0.198 was measured by the non-contact method. From the above, it can be concluded that the actual surface quality after turning with the CBN tool is better than that shown by the touch measurement.

If we compare the results obtained with similar work in the field in the last 5 years, we find that there are more scientific papers dealing with turning hardened steels with cubic boron nitride.

In the work of Nur, R et al. [26], the authors used similar cutting conditions, e.g., cutting speed $v_c$ 153 m·min$^{-1}$, feed $f$ 0.05 mm·rev$^{-1}$ and tool tip radius 0.8 mm. They address the reduction of energy consumption in metal machining. The premise is to determine the energy consumption by direct or indirect measurement. The manufacturing process under consideration is the finishing turning of mainly hardened steels. In this paper, it is proposed to use the measured cutting forces to calculate the power consumption in metal finishing turning, where the depth of cut is usually less than the cutting tool tip radius. Patel, V.D. and Gandhi, A.H. [27] addressed the use of AISI D2 steel as a material for bearing raceways, forming dies, punches, forming rolls, etc. Experiments on the finishing turning of hardened AISI D2 steel using cubic boron nitride (CBN) tools were carried out with different combinations of cutting speed (80, 116 and 152 m·min$^{-1}$), feed rate (0.04, 0.12 and 0.2 mm·rev$^{-1}$) and tool tip radius (0.4, 0.8 and 1.2 mm) using a full-factorial design for the experiments. Based on the experimental results, an empirical model of cutting forces as a function of cutting parameters (i.e., cutting speed, feed rate, and tool tip radius) has been developed. This model is based on the DOE experiment and is not due to an empiric theory. The results are comparable, but not quite universal, for the full range of $a_p$ and $f$ as in the equations in this paper.

## 5. Conclusions

This article addresses two areas of assessing the results of the hard turning of hardened materials. The first area focuses on the cutting force experiment, the results of which were used to develop a structural equation for the cutting force by theoretical mathematical derivation. The second area addresses the evaluation of the surface quality by the contact method, which is available in common industrial plants, and the verification of the measured values by the non-contact method, which in the case of laser-based measuring technology is more a matter for a scientific department.

The individual partial results of the experiment confirm the following facts:

- The construction of an empirical structural equation allows the prediction of the cutting force in hard turning finishing under similar machining conditions;
- The use of inserts from another manufacturer achieved similar surface quality results;
- The machined surface could be accepted for the four selected samples if Ra = 0.3 was taken as the cut-off value. If this cut-off value was extended to Ra = 0.4, all samples would pass;

- Comparative measurements of surface roughness values by the non-contact method showed similar or better results than the contact method.

Another important result is the confirmation of the minimal influence of the cutting speed on the cutting force, which subsequently allows simplification of the mathematical calculation of the structural equation. This article respects the needs of industry practice, where touch measurement is common. However, if verification by the non-contact method is needed, most companies will turn to a scientific institute that has this method. The same is true for the determination of cutting forces, where most industrial companies do not have any measuring equipment and can use the structural equation above, if necessary, as long as the cutting conditions are within the specified range. Of course, there is also the possibility of contacting a scientific institute.

**Author Contributions:** Conceptualization, K.O.; methodology, J.Z.; software, P.S. and J.C. All authors have read and agreed to the published version of the manuscript.

**Funding:** This research was funded by the project FV 40225 "Research and development of advanced manufacturing technology for the production of thrust spherical roller bearings with higher utility value", program TRIO, Ministry of Industry and Trade of the Czech Republic.

**Institutional Review Board Statement:** Not applicable.

**Data Availability Statement:** Not applicable.

**Acknowledgments:** Samples of semi-finished hardened material were provided by the firm ZKL.

**Conflicts of Interest:** The funders had no role in the design of the study; in the collection, analyses or interpretation of data; in the writing of the manuscript, or in the decision to publish the results. The authors declare no conflict of interest.

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
