# Peer review of "Cutting Force When Machining Hardened Steel and the Surface Roughness Achieved"

_applsci, doi:10.3390/app122211526_

Round 1

Reviewer 1 Report (Previous Reviewer 1)

The comments were not properly introduced into the text by the authors. The article title and introduction are inconsistent with the research. The introduction is not focused on the research in progress and has no scientific value. There is no information regarding the results of other researchers related to cutting forces and surface roughness of hardened steels. Lines 95-113 describe issues related to the research methodology. The test methodology contains a lot of unnecessary information, e.g. details of equipment or Fig. 1-3. The test methodology should only contain information regarding the tests performed. This chapter is too long. The research results are not described properly. E.g., the authors write: Comparison of Ra and Rz values in the touch and non-contact method shows almost the same values for samples 5 and 6, while better values were measured in samples 8 and 12 by the non-contact method. In the following Figures 15 to 22, detailed representations of the progression of the individual variables for the samples are given. On what basis was it concluded that better values measured had the non-contact method? The 2D and 3D images are left uncommented.

Author Response

Dear Reviewer,

Thank you for your review and comments. The following corrections have been made:

The comments were not properly introduced into the text by the authors. The article title and introduction are inconsistent with the research.

The title of the article has already been changed to better reflect the content of the article.

The introduction is not focused on the research in progress and has no scientific value. There is no information regarding the results of other researchers related to cutting forces and surface roughness of hardened steels.

The introduction has been substantially expanded to include information about the work of other researchers, especially from the last 5 years. The text of the introduction has been completely rewritten, and the different parts have been linked in a way that the introduction better describes the content of the article.

Lines 95-113 describe issues related to the research methodology. The test methodology contains a lot of unnecessary information, e.g. details of equipment or Fig. 1-3. The test methodology should only contain information regarding the tests performed. This chapter is too long.

Expansion of the device information and refinement of Figure 3 was requested by another reviewer. Figure 3 has been modified and simplified.

The research results are not described properly. E.g., the authors write: Comparison of Ra and Rz values in the touch and non-contact method shows almost the same values for samples 5 and 6, while better values were measured in samples 8 and 12 by the non-contact method. In the following Figures 15 to 22, detailed representations of the progression of the individual variables for the samples are given. On what basis was it concluded that better values measured had the non-contact method?

The conclusion is that different values were measured for samples 8 and 12 using the touch method and different values using the non-touch method. This is merely stated with the assumption that the non-contact method (Alicona roughness meter) is more accurate than the touch method.

The 2D and 3D images are left uncommented.

The images have been rearranged for clarity and briefly annotated.

Yours sincerely

K.Osička

Reviewer 2 Report (New Reviewer)

The article presents interesting research results and their analysis. The conventional processing of hardened steel should still be improved to inrease the process performance and reduce the cutting tool wear. The topic of the article is the most current. The presented research material in the article, however, requires corrections to improve the quality of work. Below are detailed suggestions and comments:

1. Please put information about the purpose of the research and their novelties in the abstract. Also, an abstract, according to my opinion, should focus more on the description of the tests carried out their purpose, novelty and the most important results obtained than the description of the process and phenomena occurring in it.

2. At the end of the Introduction, there should be a fragment describing what is studied in the article, what method, what impact of parameters on resulting factors, which was attempted to achieve. Introduction also requires refinement to create a uniform whole. It should also contain authors' comments to the cited results of the research of other researchers.

3. Please describe on Figure 1 in the pictures that applies to the sample before processing and which after processing.

4. Most of the figures presented in the article is too large, I suggest you reduce them. They unnecessarily increase the number of pages of the article. Also, many charts and photos require improvement in quality.

5. Information such as: formulas used to calculate resulting factors, statistical analysis results, tables with determined standard deviation, should be included in the chapter "Materials and Methods".

6. Please also indicate research results from more scientific articles from the last 5 years.

Author Response

Dear Reviewer,

Thank you for your review and comments. The following corrections have been made:

  1. Please put information about the purpose of the research and their novelties in the abstract. Also, an abstract, according to my opinion, should focus more on the description of the tests carried out their purpose, novelty and the most important results obtained than the description of the process and phenomena occurring in it.

The abstract has been expanded to include the requested information

  1. At the end of the Introduction, there should be a fragment describing what is studied in the article, what method, what impact of parameters on resulting factors, which was attempted to achieve. Introduction also requires refinement to create a uniform whole. It should also contain authors' comments to the cited results of the research of other researchers.

The introduction has been substantially expanded to include information about the work of other researchers, especially from the last 5 years. The text of the introduction has been completely rewritten, and the different parts have been linked in a way that the introduction better describes the content of the article.

  1. Please describe on Figure 1 in the pictures that applies to the sample before processing and which after processing.

The position of the machined sample and the blank was specified

  1. Most of the figures presented in the article is too large, I suggest you reduce them. They unnecessarily increase the number of pages of the article. Also, many charts and photos require improvement in quality.

The images were reduced in size and the quality was improved.

  1. Information such as: formulas used to calculate resulting factors, statistical analysis results, tables with determined standard deviation, should be included in the chapter "Materials and Methods".

The derivation of the structural equation, the results of the statistics, the tables with the determined standard deviation are related to the measured data and are more clear in this way.

  1. Please also indicate research results from more scientific articles from the last 5 years.

Research results from the last 5 years have been added to the discussion. The introduction itself has been substantially revised.

Yours sincerely

K.Osička

Reviewer 3 Report (New Reviewer)

The manuscript established an experienced equation to predict the cutting force within a certain range by cutting 100 Cr6 the steel with PCBN tools; the results of contact and non-contact methods to measure surface roughness were compared, and the surface roughness measured with both methods was similar. However, it still exists some shortcomings.

1.Line 294Why choose the data of group 5,6,8,12

2.Line 314: Sample10 should be sample12?

3.Lines 331and 335: Table 4 should be Table5?

4. Line 360: Is  of Fig.12 deduced? What is the derivation process?

Author Response

Dear Reviewer,

Thank you for your review and comments. The following corrections have been made:

1.Line 294:Why choose the data of group 5,6,8,12?

The data marked in green were selected for measurement by the non-contact method for the following reasons. Samples no. 5 and no. 6 have the best surface quality result measured by the touch method. In contrast, sample no. 8 has the worst surface quality result.  Sample no. 12 was selected because of the maximum cutting conditions used.

2.Line 314: Sample10 should be sample12?

Yes, it has been corrected.

3.Lines 331and 335: Table 4 should be Table5?

Yes, it has been corrected. An additional table 6 has been added and the others have been renumbered.

  1. Line 360: Is  of Fig.12 deduced? What is the derivation process?

The procedure given in equations 3.3 to 3.5. After logarithmization, xFc can be defined as the directive of the line tg α.

Yours sincerely

K.Osička

Round 2

Reviewer 1 Report (Previous Reviewer 1)

The issues addressed in the paper titled. "Cutting force and surface quality when machining hardened steel” are important for industrial application. However, the work contains many aspects that need significant improvement.

The title of the paper is incorrectly worded. Please explain how "surface quality when machining" can be studied. It should also indicate the name of the material studied.

The introduction is far too long. It should include a brief introduction to the topic, recent data from similar research work, and what is the subject of your own research and why such research was undertaken.

Surface quality cannot be measured. Surface roughness, surface topography, etc. can be measured.

The detailed data of the domiarization equipment provided by the equipment manufacturers is unnecessary. It is only necessary to provide information on the research being conducted.

Why, having such advanced equipment, Ra and Rz parameters were measured, and for what purpose contact measurements were also made. Why was 50x magnification used for Alicon measurements? It may be too high to properly evaluate the machined surfaces.

Table 5 gives the "resulting force F". How was the result determined.

The authors write "Figure 13 shows a contour plot of the effect of ap and f on the cutting force Fc. Thanks to the contour representation, it is possible to observe the relationships and optimize the process, also the uniform distribution of the dependence for both f and ap can be seen." - If an influence diagram is presented, the analysis should show the effect of ap and f on Fc. For which parameters, for example, it is possible to obtain a lower force Fc.

Figure16 Screen of the roughness meter after the measurement. - this is a scientific paper, it makes no sense to show screen of the measurement results. It does not add anything important to the work.

The authors write "The data marked in green were selected for measurement by the non-contact method for the following reasons. Samples no.5 and no.6 have the best surface quality result measured by the touch method. In contrast, sample no.8 has the worst surface quality result. Sample no.12 was selected because of the maximum cutting conditions used. The other samples were not measured by non-contact for capacity reasons. For the surface roughness parameter, the average arithmetic deviation of the profile Ra, the standard deviation values are given in Table 9 below." Such wording should not take place and is incorrect: "the best surface quality result measured"; "the maximum cutting conditions used"; "The other samples were not measured by non-contact for capacity reasons.

The authors write: "In general, the resulting surface finish depends mainly on the feed rate f, the lower the feed rate f, the lower the surface finish. The average values of the arithmetic deviation of the profile under consideration, Ra, are between 0.3 and 0.4 μm." - what is the definition of quality? Can quality be assessed by the parameter f? Instead, it has long been known that f affects surface roughness.

The authors can: "In general, based on the results obtained from the contact measurements, it can be stated that out of 12 machined samples with different combinations of cutting conditions, 11 samples were below Ra = 0.4 μm on average during the measurements. It is difficult to reach this value with conventional finishing tools made of materials other than CNB. It follows that the CNB insert material is able to replace finishing operations such as grinding in certain cases." - Please prove this thesis.

E.g. in Table 9 - there is no description of the results in post 2 and 4.

Photo captions are misplaced. E.g.: Fig. 21 or 22.

Photos are of poor quality, e.g. fig 4.

There are many grammatical and linguistic errors in the text.

To summarize. The results of the study are mostly presented in an unscientific and very general way. The work contains a great deal of unnecessary information that does not contribute to the quality of the work, but only prolongs its volume. The quality of the photos is poor. Some descriptions of the photos are also inadequate.

Author Response

Open Review 1 (Round 2)

Dear Reviewer,

Thank you for your review and comments. The following corrections have been made:

The title of the paper is incorrectly worded. Please explain how "surface quality when machining" can be studied. It should also indicate the name of the material studied.

The title of the article "Cutting force and surface quality in hardened steel machining" has been changed again to "Cutting force in hardened steel machining and surface roughness achieved" to better reflect its content.

The introduction is far too long. It should include a brief introduction to the topic, recent data from similar research work, and what is the subject of your own research and why such research was undertaken.

The introduction has been expanded and substantially revised as requested in previous revisions.

Surface quality cannot be measured. Surface roughness, surface topography, etc. can be measured.

The concept of surface quality is meant in general, the individual measured variables are described in detail in the article.

The detailed data of the domiarization equipment provided by the equipment manufacturers is unnecessary. It is only necessary to provide information on the research being conducted.

Detailed data on measuring devices was requested in previous revisions, however some data on measuring devices has therefore been removed e.g. Taylor Hobson Surtronic S-128 touch tester data and Table 4 Alicona technical specification.

Why, having such advanced equipment, Ra and Rz parameters were measured, and for what purpose contact measurements were also made. Why was 50x magnification used for Alicon measurements? It may be too high to properly evaluate the machined surfaces.

The 50x magnification setting corresponds to the expected values of the Ra and Rz parameters.

Table 5 gives the "resulting force F". How was the result determined.

"The resulting force F" now in Table 4 is one of the outputs on the Kistler dynamometer and corresponds to the calculation relation.

F= ((Fc)2+(Fp)2+(Ff)2)0.5

The authors write "Figure 13 shows a contour plot of the effect of ap and f on the cutting force Fc. Thanks to the contour representation, it is possible to observe the relationships and optimize the process, also the uniform distribution of the dependence for both f and ap can be seen." - If an influence diagram is presented, the analysis should show the effect of ap and f on Fc. For which parameters, for example, it is possible to obtain a lower force Fc.

The individual coloured fields show where the cutting force Fc will be at specific values of ap and f

Figure16 Screen of the roughness meter after the measurement. - this is a scientific paper, it makes no sense to show screen of the measurement results. It does not add anything important to the work.

Fig.16 and the references in the text to it have been removed. Next, Figures 5,6,7 and all references in the text have been removed. Other pictures have been renumbered.

The authors write "The data marked in green were selected for measurement by the non-contact method for the following reasons. Samples no.5 and no.6 have the best surface quality result measured by the touch method. In contrast, sample no.8 has the worst surface quality result. Sample no.12 was selected because of the maximum cutting conditions used. The other samples were not measured by non-contact for capacity reasons. For the surface roughness parameter, the average arithmetic deviation of the profile Ra, the standard deviation values are given in Table 9 below." Such wording should not take place and is incorrect: "the best surface quality result measured"; "the maximum cutting conditions used"; "The other samples were not measured by non-contact for capacity reasons."

 More detailed information on why these samples were marked green was requested in the previous revision. The wording has been corrected subsequently:

 "Samples 5 and 6 have the best value for Ra and Rz, sample 8 has the worst value when measured by the touch method. For sample 12, the highest values of Ra and Rz were used.”

The authors write: "In general, the resulting surface finish depends mainly on the feed rate f, the lower the feed rate f, the lower the surface finish. The average values of the arithmetic deviation of the profile under consideration, Ra, are between 0.3 and 0.4 μm." - what is the definition of quality? Can quality be assessed by the parameter f? Instead, it has long been known that f affects surface roughness.

One of the definitions of quality is the complex concept of "Surface Integrity" which includes the properties and characteristics of the surface (surface area and surface layer) produced as an output of the machining process. During the machining process, changes occur on the surface and in the surface layer that may result in plastic deformation, structural changes, hardening, residual stresses, hardness changes, etc.

The authors can: "In general, based on the results obtained from the contact measurements, it can be stated that out of 12 machined samples with different combinations of cutting conditions, 11 samples were below Ra = 0.4 μm on average during the measurements. It is difficult to reach this value with conventional finishing tools made of materials other than CNB. It follows that the CNB insert material is able to replace finishing operations such as grinding in certain cases." - Please prove this thesis.

The above technology of machining hardened steels into bearing components was tested using CBN inserts of different types and we achieved better results of the Ra parameter (0.4 μm) and at the same time the state of compressive stress in the surface layer. These are mainly finishing operations on shaped rotating components. However, these results and the corresponding cutting conditions are not the subject of this paper.

E.g. in Table 9 - there is no description of the results in post 2 and 4.

Table 9 (now 8) was introduced following a request from a previous revision to add a standard deviation Ra.

Photo captions are misplaced. E.g.: Fig. 21 or 22.

The position of the captions of Figures 20, 21 ( after renumbering Figures 17,18) has been corrected.

Photos are of poor quality, e.g. fig 4.

The photograph in Figure 4 has been replaced.

There are many grammatical and linguistic errors in the text.

To summarize. The results of the study are mostly presented in an unscientific and very general way. The work contains a great deal of unnecessary information that does not contribute to the quality of the work, but only prolongs its volume. The quality of the photos is poor. Some descriptions of the photos are also inadequate.

Summary. All specific comments were accepted and questions answered. The total length of the paper has been reduced from 23 pages to 21 pages.

Yours sincerely

K.Osička

Round 3

Reviewer 1 Report (Previous Reviewer 1)

Thank you for the answers provided by the authors of the article. They are satisfactory and comprehensive. In my opinion, the article can be published in the Journal Applied Sciences.

This manuscript is a resubmission of an earlier submission. The following is a list of the peer review reports and author responses from that submission.

Round 1

Reviewer 1 Report

The paper entitled "Cutting force when machining hardened steels" presents a study of the cutting forces and surface roughness of machined 100Cr6 (1.3505) bearing steel.

The article needs thorough revision, which is described below.

The title of the article should be worded differently. In addition to cutting forces, the surface roughness after turning 100Cr6 steel is analyzed.

Abstract:

- Determine the grade of hardened steel.

- No details of the tests.

- Include at the end of the abstract what was determined, what test results were obtained.

Keywords:

- Should include important words: surface roughness, cutting force, turning.

Chapter 1

- The review is insufficient. It is necessary to present the latest scientific developments (results of similar works) that deal with cutting forces and shaping of the machined surface of various hardened steels after the turning process.

Chapter 2

- The authors write: „Component material the rolling element is 100 Cr6 according to ISO.” - please indicate the exact standard.

Chapter 2.2

It contains a small amount of data.

- Please specify the exact model of the Kistler dynamometer and the sampling rate when taking the measurements.

- Please give the correct names for each cutting force at first use: Fc, Ff, Fp

- The authors write: „The machine used for the experiment was the SV 18 RD, which is a lathe that has been verified for rigidity. The machine also has a considerable range of cutting speeds that can be continuously controlled.” How was the rigidity of the machine checked? What is the cutting speed range? Manufacturers usually specify the maximum spindle speed. What is the motor power of the machine? These aspects are very important.

- Please provide the full name of the lathe.

- Whether a CNC or conventional lathe was used.

- The authors should provide the exact geometry of the cutting tools, which is extremely important when turning hard materials, namely: tool cutting edge angle, rake angle, tool clearance angle, nose radius, etc. The information given in Table 2 (exceptnose radius) is unnecessary. Why are the f or ap ranges given? Since completely different ones were used in the study. Completely different parameters were used in the study.

- The authors write: „The machining was carried out without process fluid and due to the geometry of the cutting insert (without chip sealer) a continuous segmented chip is realized, as shown in Figure 3.”  The authors should prove this conclusion and show a close-up photo of the chips (without the machine tool). Nothing can be seen in Fig. 3. Please justify what effect such a chip shape has on machining. At what parameters such a chip was obtained? For all combinations? On the other hand, what effect does this conclusion have on cutting forces and surface roughness?

Chapter 2.4

- In my opinion there is no point in giving technical parameters of testers. The length of an elementary (measuring) section and the number of measuring sections should be given. What are the advantages and disadvantages of contact measurements.

Chapter 2.5

- What measurement technique is used by Alicona Infinite Focus G5. Please describe its advantages and disadvantages.

- In what program were the results processed?

- The authors write: „used 50x lens; filter Lc [μm] 800; profile measurement path length [mm] 4”. The authors use high magnification (50x) - so what was the measurement area (for 3D measurements we cannot write about path length. The measurement area should be given. On the other hand, isn't 4 mm too small to evaluate the surface roughness?

Chapter 3.1

What do the green rectangles in the tables mean?

Chapter 3.2

The authors provide a number of formulas, please provide their literature references.

Chapter 3.3.2

For better readability, it is recommended to present the test parameters in a table.

- 2D and 3D images of the machined surface, the surface roughness profiles are of poor quality. 2D and 3D images should be enlarged and more readable. Scale is not readable with 3D images.

Chapter 4.

This chapter should include a scientific discussion. Currently, it presents only the research conclusions alone, without reference to other current scientific work. Therefore, it requires significant changes.

In my opinion the article looks more like a technical report than a research article.

Other aspects to be explained:

Please demonstrate what is new in the paper compared to other scientific papers.

In my opinion it does not make sense to present and compare the results of two measuring devices. The authors should not measure 2D surface roughness parameters since they decide to show 3D surfaces. So it does not make sense to compare contact and non-contact devices. Current scientific research mainly uses 3D surface roughness parameters, namely Sa, Sz, etc.

The authors give the feed rate in [mm]. This is not true. The unit of feed rate for turning is mm/rev. The entire work should be checked and corrected.

Was the cutting blade replaced after each sample?

Providing photos 1 - 7 (poor quality) adds nothing. It would be more beneficial to use them and present the experimental plan in graphical form, and remove the photos. It would increase the quality of the article.

Parameters such as f, vc, ap, Fc,Ff, Fp... etc. should be written in italics.

The first time an abbreviation is used it should be explained.

References should include more recent items from the last 3 years. On the other hand, the quantity and quality of scientific papers presented in this article is low.

Reviewer 2 Report

 The following questions should be considered seriously.
1.In Section 2.1, the authors say that “Unlike previous experiments with bearing rings, the SP280 SY machine was not used because of the practically impossible placement of the Kistler measuring system on this machine, namely the placement of the measuring probes on the vertical slide of the machine.”, The authors attempt to explain the refusal to use SP280 SY machine due to difficulty in installation of Kistler 9257B. Figure 3 shows the complete instrument used during the experiment. It is suggested that the authors give a clear diagram of installation of Kistler 9257B on SV 18 RD machine.
2.Table 2 shows the dimensional specification of the insert used. While, ap, f and vc are cutting parameters, not the dimensional specification of the insert.
3.Table 3 shows the cutting conditions of the experiment. Please give a brief discussion on the basis for determining the range of cutting parameters.
4.During longitudinal turning, the force components are measured in three directions according to the coordinate system, i.e. Fx, Fy and Fz in the x, y and z axes. While in Table 4 Average values of individual cutting forces, Fc, Ff, Fp is given. How do forces transform in different coordinate systems?

5.The quality of Figures 8,9,10,11,15,17,19,21 should be improved.

Reviewer 3 Report

The manuscript entitled “Cutting force when machining hardened steels " authored by Karel Osička et al. attempts to assess the cutting forces and surface quality of hardened steel after machining with boron nitride tools. The title of the article is very generic and the introduction is average and it is mostly citing the authors' previous work without adequate reason. There is no real literature review and the purpose of citing some articles is also not clear.

In my opinion, this manuscript is limited to very standard machining experiments widely available in the literature and without any clear innovation content. Thus, I rejected this paper in its current form.